# Evaluating Food Procurement against the EAT-*Lancet* Planetary Health Diet in a Sample of U.S. Universities

**DOI:** 10.3390/ijerph21070945

**Published:** 2024-07-19

**Authors:** Jaclyn Bertoldo, Abby Fammartino, Sophie Egan, Roni A. Neff, Rebecca Grekin, Julia A. Wolfson

**Affiliations:** 1Department of International Health, Johns Hopkins University Bloomberg School of Public Health, Baltimore, MD 21205, USA; jwolfso7@jhu.edu; 2Strategic Initiatives Group, Culinary Institute of America, Hyde Park, NY 12538, USA; abby.fammartino@culinary.edu; 3R&DE Stanford Food Institute, Stanford University, Stanford, CA 94305, USA; smegan@stanford.edu; 4Department of Environmental Health and Engineering, Johns Hopkins University Bloomberg School of Public Health, Baltimore, MD 21205, USA; rneff1@jhu.edu; 5Department of Health Policy and Management, Johns Hopkins University Bloomberg School of Public Health, Baltimore, MD 21205, USA; 6Department of Energy Science and Engineering, Stanford University, Stanford, CA 94305, USA; rgrekin@stanford.edu

**Keywords:** nutrition, healthy diets, food systems, food procurement, planetary health, climate change, sustainability

## Abstract

Aligning institutional food procurement with planetary health targets offers opportunities to improve nutrition and reduce food-related greenhouse gas (GHG) emissions. This study compared foods procured by 19 university dining programs in the U.S. in 2022 with the EAT-*Lancet* planetary health diet. Each university’s procurement was then modeled to align with the EAT-*Lancet* planetary health diet, and changes to Healthy Eating Index (HEI) scores and GHG emissions were evaluated. For a subset of universities that provided cost data, changes in annual total food costs were also estimated. Universities in this study exceeded EAT-*Lancet* planetary health targets for beef ($\overset{-}{x}$ = 657% of target), pork ($\overset{-}{x}$ = 587%), poultry ($\overset{-}{x}$ = 379%), and eggs ($\overset{-}{x}$ = 293%). All universities failed to achieve planetary health targets for legumes and nuts ($\overset{-}{x}$ = 39% of the target) and vegetables ($\overset{-}{x}$ = 68%). Aligning food procurement with the planetary health diet would result in an estimated average 46.1% reduction in GHG emissions and a 19.7 point increase in HEI scores. Universities that provided cost data saw an average 9.7% reduction in food costs in the EAT-*Lancet*-aligned scenario. The procurement metrics assessed in this study can help university dining programs and other institutional food service organizations set goals and monitor progress toward planetary health targets.

## 1. Introduction

Food systems are inextricably connected to human and planetary health [1]. Food is at the nexus of climate change, environmental degradation, obesity, and malnutrition, which bear substantial global burdens and public health implications [2,3,4]. For these reasons, it is of increasing interest to researchers, practitioners, and policymakers to identify ways to concurrently evaluate and address the impacts of food on the environment and human nutrition. In 2019, a group of global experts spanning the fields of human health, agriculture, political science, and environmental sustainability published *Food in the Anthropocene: the EAT-Lancet Commission on healthy diets from sustainable food systems*, outlining the first global framework and reference diet for optimizing human health within planetary boundaries [1]. The EAT-*Lancet* planetary health diet aims to improve dietary health and reduce food-related greenhouse gas (GHG) emissions, among other environmental impacts, by promoting a plant-forward diet that emphasizes whole and minimally processed grains, vegetables, fruit, legumes, and nuts while limiting meat and dairy [1]. Plant-forward diets are not vegan, and the planetary health diet accommodates an omnivorous eating pattern that is intended to be adaptable across different cultures and contexts. However, it recommends substantial reductions in meat consumption, particularly for affluent countries that are heavy consumers. The EAT-*Lancet* Commission assessed global and regional food consumption in 2016 against the planetary health diet and found that North America exceeded the planetary health targets for beef, pork, poultry, dairy, eggs, and potatoes while failing to meet planetary health targets for fish, legumes, nuts, whole grains, vegetables, and fruit. A small number of studies have subsequently measured the alignment of individual diets [5], recipes [6], and school menus [7] with the EAT-*Lancet* planetary health diet. However, no research to date has attempted to evaluate the alignment of institutional food procurement with the EAT-*Lancet* planetary health diet.

Institutional food procurement, which encompasses the purchasing activities of foodservice operations serving institutions such as schools, hospitals, and universities [8], has been identified as an important sector for influencing healthy and sustainable food production and consumption [9,10]. Aligning institutional food procurement with the EAT-*Lancet* planetary health diet presents a unique opportunity to leverage institutional purchasing power to incentivize food suppliers and manufacturers to produce more plant-forward food options that are nutritious and have lower environmental impacts. Institutional food programs are also in a position to construct food environments that could promote healthy and sustainable food choices to the large number of diners they serve each day. For example, prior work has indicated that campus food environments with greater accessibility to healthy food options can beneficially impact students’ dietary intake [11]. The EAT-*Lancet* framework can be useful in guiding institutional food procurement by offering science-based targets for the contribution of different food categories within the procurement portfolio to optimally promote the nutritional quality of food purchases while reducing food-related GHG emissions.

Shifting institutional procurement towards the planetary health diet is a strategy set forth by the C40 Cities Climate Leadership Group, a global network of nearly 100 cities that are united in efforts to confront climate change [12]. A subset of 16 cities in this network, including the U.S. cities of New York City and Los Angeles, have committed to align their city’s institutional food procurement with the planetary health diet by 2030 [13]. However, mechanisms for evaluating and measuring progress towards this target are currently lacking [14]. Strategies are urgently needed to measure the alignment of food procurement with the EAT-*Lancet* planetary health diet and to understand gaps between current food purchasing patterns and the EAT-*Lancet* targets for different categories of foods. This type of assessment could also be useful for evaluating the potential benefits of aligning procurement with EAT-*Lancet* guidelines across both nutrition and climate metrics. For example, the Healthy Eating Index (HEI) is a commonly applied metric that can be used to evaluate nutritional quality across a set of foods [15] and measuring food-related GHG emissions is a common strategy for evaluating the climate impact of institutional food purchases [9,16,17]. However, these metrics have not yet been applied together to evaluate the nutritional and environmental impacts of shifting institutional food procurement to align with the EAT-*Lancet* planetary health diet. Additionally, the potential cost implications of these shifts have not been evaluated.

Universities are an example of an institutional setting in which evaluating the alignment of procurement practices with the EAT-*Lancet* planetary health diet would be valuable. Emerging adulthood is an important life stage for developing healthy eating habits, as many young adults are at increased risk for weight gain and poor nutrition during this transformational period, and dietary behaviors learned during this time can carry forward into adulthood [18]. Given the direct interplay between food procurement and the campus food environment, university dining programs are increasingly focused on advancing healthy and sustainable food options in recognition of the opportunity to positively influence the dietary trajectory of their students [19]. For example, the University of California system has committed to a Healthy Campus initiative that includes system-wide healthy food purchasing and vending policies [20]. Additionally, a growing number of universities are committing to reducing total GHG emissions associated with their food purchases in efforts to promote environmental sustainability [17,21]. Evaluating the alignment of university food purchases with the EAT-*Lancet* planetary health diet can help advance ongoing efforts in this sector to promote healthy and sustainable food programs across university campuses while providing a model for other types of institutions to assess their food procurement patterns in the context of both nutrition and environmental impacts. This study aimed to analyze foods procured by university dining programs over a 1-year period to evaluate alignment with the EAT-*Lancet* planetary health diet and assess the potential impacts of aligning procurement with EAT-Lancet guidelines on nutrition quality, GHG emissions and food costs.

## 2. Materials and Methods

Food purchases were categorized into the planetary health diet food categories, and the contribution of each food category to the total product mix was compared against planetary health targets defined in the EAT-*Lancet* planetary health diet [1]. Each university’s procurement portfolio was then modeled to align with the EAT-*Lancet* planetary health diet, and changes in food-related GHG emissions, nutrition quality (HEI scores), and food costs were evaluated. A process diagram outlining the materials and methods in this study can be found in the Appendix A (Figure A1). This study was approved by the Johns Hopkins Bloomberg School of Public Health Institutional Review Board.

### 2.1. Recruitment

University dining programs were recruited to participate in this study through the Menus of Change University Research Collaborative (MCURC) member database. The MCURC is an international network of colleges and universities committed to advancing plant-forward food programs that promote healthy, sustainable, and delicious food choices [19]. Its membership includes university administrators, dining directors, chefs, nutrition and sustainability managers, and academic faculty representing 63 colleges and universities at the time of this study, primarily based in the U.S. but also in the United Kingdom, Spain, France, and Singapore. In February 2023, a description of the study and an invitation to participate was sent via email to all MCURC members who were affiliated with a university dining program and had previously opted to receive MCURC communications, representing a total of 43 university dining programs. Ultimately, 20 university dining programs agreed to participate in the study, but one was ultimately excluded from the analysis because it was located outside of the United States, resulting in an analytic sample composed of 19 university dining programs.

### 2.2. Data Sources

Participating dining programs were asked to run retrospective procurement reports that included the name, pack size, and total weight or volume of all food and beverage items that were procured over a one-year period from 1 January to 31 December 2022. Optionally, universities could provide data on the cost of each item purchased over this period. University dining programs submitted their procurement data via email in an Excel spreadsheet. Each dining program submitted one year’s worth of procurement data from 2022, except one university that was only able to provide six months of purchase data. This university’s data was extrapolated to a full year’s worth of purchases for analysis. Of these university dining programs, 8 provided data on food costs. University descriptive characteristics, including size, region, public or private status, % students identifying as female, % students identifying as a racial/ethnic minority, and % Pell grant eligible students, were determined from online sources [22,23]. 

### 2.3. Food Categorization

Food and beverage names provided in the procurement reports were used to categorize food items into the EAT-*Lancet* planetary health diet food groups for analysis. The procurement data was first run through the food categorization function of the Automated Scope 3 Tool for Tracking Emissions from Food (TASTE Food), a Python-based procurement analysis tool that was previously developed and validated with university food procurement data [24,25]. This tool uses the description of each food item to match the food to at least one and up to three different food categories defined by the SIMAP emissions accounting platform [26], which has a strong alignment with the EAT-*Lancet* planetary health diet food categories. Once each food item was categorized with this tool, researchers manually verified whether food items were assigned to the correct food category (or categories), re-assigning items and adding supplemental categories as needed. If an item’s name was insufficiently descriptive to assign it to a food category, the researchers consulted with dining program staff from the institution that provided the data to determine the appropriate food category. The full list of food categories, descriptions, and reference foods for analysis can be found in the Appendix A (Table A1). Since the assignment of food categories was based on the names of food items and was limited to a maximum of three food groups, mixed and prepared items with many different ingredients were assigned to food groups based on the three ingredients assumed to make up the greatest proportion of their weight. For example, an item described as “beef and cheese lasagna” was assigned to the beef, dairy, and grains categories, and its weight was evenly distributed across each of these categories. For sugar-sweetened grain products, dairy, and beverages, the percentage of the product’s weight that was comprised of sugar was approximated using USDA nutritional data from reference foods and attributed to the sugar category, while the remaining weight was distributed across other relevant food categories (i.e., grains, dairy, and liquids). Since the dietary components described in the EAT-*Lancet* planetary health diet are based on the raw, edible weight, food descriptions were also used to infer whether the item’s weight should be adjusted to account for the bone weight, moisture loss or gain due to cooking, inedible components removed in processing, or concentration or dilution of beverages [27]. For example, if a bean product was described as “canned” in the item name, the total weight was adjusted to account for draining the canning liquid and reverting cooked beans back to dry weight. Additionally, the EAT-*Lancet* dairy category is based on whole milk equivalents, so all dairy products were converted to their equivalent weight in milk based on the USDA MyPlate serving equivalents [28]. All weights and volumes were converted to kilograms for analysis.

### 2.4. Comparing Food Purchases to EAT-Lancet Recommendations

Researchers calculated the weight of foods purchased for each EAT-*Lancet* food category and then summed the weight of all food categories to determine the total weight of each university’s procurement portfolio. To compare the university procurement portfolios with the EAT-*Lancet* planetary health diet, the weights of each food category were first divided by the total weight of foods purchased to determine the percent contribution of each food category in the procurement portfolio. These percentages were then divided by the planetary health targets for each food category (i.e., the “ideal” percent contribution of each food type to the total mix of foods as defined in the planetary health diet) to determine the deviation above or below the planetary health target for each food category. This approach mirrors the EAT-*Lancet* Commission’s comparison of current dietary patterns in the U.S. and globally with the planetary health diet [1]. All analysis was conducted in Excel, version 16.80 [29].

### 2.5. Estimating Food-Related GHG Emissions

To estimate the GHG emissions associated with each university’s procurement portfolio, this study utilized the methods and GHG emissions factors from the World Resources Institute Cool Food Calculator, a validated tool for calculating upstream GHG emissions related to institutional food purchases from the cradle to the point of purchase [16]. The Cool Food Calculator has been previously utilized to estimate GHG emissions from food procurement across dozens of universities and other institutional procurers, including several dining programs in the MCURC [30]. As described in the Cool Food Calculator technical note, default GHG emissions factors were derived from a life-cycle meta-analysis conducted by Poore and Nemecek in 2018 [16,31]. GHG emission factors for North American agricultural supply chains include the average estimated GHG emissions from energy use on the farm, enteric methane from ruminant animals, animal waste, production and application of fertilizers, methane associated with rice production, transport of food and animal feed, food processing, packaging, and food losses from the cradle to the point of purchase [16,31]. Food-related GHG emissions were calculated by multiplying the total weight of each food category in kilograms by the associated GHG emissions factor for that category in the Cool Food Calculator. Composite GHG emissions factors were calculated for the grain, nut and seed, and oil categories by averaging the Cool Food Calculator GHG emissions factors for the food types within these categories that were most common in the university procurement data. For example, the GHG emissions factors for wheat and rice were averaged to create a composite GHG emissions factor for the grains category. Liquids that did not align with an EAT-*Lancet* reference food category, such as water, vinegars, and some condiments, were excluded from the analysis. GHG emissions for sugar-sweetened beverages were estimated using the USDA nutritional profile for soda [32] to isolate the weight of sugar from the total weight of these beverages, which was multiplied by the Cool Food Calculator GHG emissions factor for sugar and sweeteners. Fruit juice GHG emissions were calculated using citrus fruit as the reference food category and estimating the conversion of juice to whole fruit using USDA food yield estimations [27]. Spices, coffee, and tea were also omitted from the GHG emissions calculations of the dataset as these categories are also not represented in the EAT-*Lancet* planetary health diet. All other food categories were mapped directly to a GHG emissions factor (or composite of GHG emissions factors) in the Cool Foods Calculator to estimate GHG emissions from that category (Table 1). GHG emissions are reported in standard carbon dioxide equivalents (CO_2_e) based on the global warming potential of different greenhouse gasses over a 100-year time horizon (GWP100) [16].

### 2.6. Calculating Healthy Eating Index (HEI)

To evaluate the nutritional quality of the university procurement portfolios, each EAT-*Lancet* food category in the analysis was mapped to a reference food, or in certain cases, a composite of reference foods, from USDA’s FoodData Central database [32]. When appropriate, EAT-*Lancet* food categories were mapped to the same USDA reference foods used in the development of the EAT-*Lancet* planetary health diet [1]. Composite reference data was calculated for the refined grains, whole grains, vegetables, dark green vegetables, red and orange vegetables, fruits, nuts, and dairy alternatives categories by identifying the two or three most common items within each category and averaging the nutritional profile evenly across those foods. Each EAT-*Lancet* food category was similarly mapped to a reference ingredient (or ingredients) in the USDA’s Food Pattern Equivalents Database in order to calculate the total servings of the dietary constituents included in the HEI [33]. A full list of reference foods from FoodData Central and the Food Pattern Equivalents Database (FPED) for each food category in the analysis can be found in the Appendix A (Table A1). Liquid foods and beverages that could not be converted into an EAT-*Lancet* food category were excluded from the analysis. As with the GHG emission calculations, spices, coffee, and tea were also excluded from analysis, with the exception of salt, which was included only in its commodity form where the weight of salt could be accurately determined. The total amount of calories, saturated fat, monounsaturated fat, polyunsaturated fat, added sugar, sodium, and servings of fruit, whole fruit, vegetables, greens and beans, whole grains, refined grains, dairy, protein foods, seafood, and plant proteins were calculated for each procurement portfolio. HEI components and total scores were then calculated according to the HEI scoring standards [33] for each university’s procurement portfolio. 

### 2.7. Cost Analysis

Of the 19 university dining programs that provided food procurement data, 8 opted to submit the costs of each item they purchased for the data collection period. For these universities, the total cost of each item was distributed evenly across the food categories assigned to that item, and the costs for each food category were then aggregated to arrive at an average cost per kilogram for each EAT-*Lancet* food category. 

### 2.8. Modeling EAT-Lancet-Aligned Procurement Scenarios

To evaluate changes in food-related GHG emissions, HEI scores, and food costs associated with aligning each university’s procurement portfolio with the EAT-*Lancet* planetary health diet, researchers held the total weight of foods purchased by each university constant and then redistributed those purchases to align with the planetary health targets for each food category. For example, if a university purchased 1 million kilograms of food and the planetary health target for beef is 0.53% of the total mix of foods by weight, then 1 million was multiplied by 0.53% to arrive at 5300 kg of beef in the EAT-*Lancet*-aligned procurement scenario. This calculation was applied to all food categories to generate a theoretical EAT-*Lancet*-aligned procurement portfolio for every university. The EAT-*Lancet* planetary health diet does not include a category for refined grains, so in the EAT-*Lancet*-aligned procurement scenario, all grains were assigned to the whole grain category. 

Once the EAT-*Lancet*-aligned procurement scenarios were generated, food-related GHG emissions, HEI scores, and food costs (when this data was available) were re-calculated for each university. The EAT-*Lancet* planetary health diet does not include a reference to the ideal amount of salt or sodium in the diet, so no target for salt was defined in the EAT-*Lancet*-aligned scenario. Therefore, for the HEI calculations, the sodium score was held constant across both procurement portfolios. All other calculations were repeated as described in the analysis of the original procurement portfolios.

## 3. Results

The characteristics of participating universities compared to all MCURC member universities can be found in Table 1. The study sample had a greater proportion of large universities (>15,000 students) compared to all MCURC universities and did not have representation from any small universities (<5000 students). The U.S. regions represented in the study sample were comparable to the broader MCURC membership, with both groups having the greatest representation from schools in the geographical North and West. Participating universities were split almost evenly between public and private institutions, with a slightly greater representation of private universities in the study sample compared to the MCURC membership. Universities contributing to the study had a similar percentage of female students and students receiving Pell grants compared to the average in these categories across all MCURC universities, and the study sample had a greater average percentage of minority students attending these institutions compared to the full MCURC membership. All participating dining programs were self-operated, meaning that their primary residential dining operations are operated by the university as opposed to being contracted to a food service management company.

The composition of university procurement portfolios compared to the planetary health targets defined in the EAT-*Lancet* planetary health diet is outlined in Table 2. All universities exceeded the EAT-*Lancet* planetary health targets for beef, pork, poultry, and eggs and failed to meet the targets for legumes and nuts, vegetables, and whole grains despite substantial variation in procurement portfolio composition between universities. 

Figure 1 demonstrates the average deviation from the planetary health target for each food category in the 19 university procurement portfolios. Aligning the purchases of the 19 universities with the EAT-*Lancet* planetary health diet would require substantial reductions in purchases of beef, pork, poultry, dairy, eggs, potatoes, and sugar and significant increases in whole grains, vegetables, fruits, and legumes and nuts purchases.

The impact of these shifts on food-related GHG emissions and HEI scores across each of the 19 universities is presented in Table 3. Dividing the GHG emissions over the total weight of foods purchased (kg) for comparison between universities, all universities would reduce food-related GHG emissions in the EAT-*Lancet*-aligned scenario, with reductions ranging from 30.5% to 58.9% across all universities. The mean GHG emission reduction across all universities was 46.1%. The plant-forward shifts in the EAT-*Lancet*-aligned scenario also improved HEI scores for all universities. Improvements to HEI scores ranged from 5.21 to 32.41 points, with an average improvement in HEI score of 19.70 points. Scores across all HEI components increased except for dairy and total protein foods, which decreased slightly, and the sodium category, which was held constant. The HEI components most responsible for the overall increase in HEI score were improvements to the whole grains, refined grains, saturated fats, and fatty acids categories. For the universities that reported food costs, aligning procurement portfolios with EAT-*Lancet* guidelines resulted in lower food costs for six of the eight universities. On average, food costs were 9.7% lower in the EAT-*Lancet*-aligned scenario. However, there was wide variability across the universities, ranging from a savings of 27.0% to an increase in costs of 3.2%.

## 4. Discussion

This study used 2022 annual food procurement data from a sample of university dining programs in the MCURC to evaluate alignment with the EAT-*Lancet* planetary health diet and explore the impacts of aligning with planetary health targets on GHG emissions, HEI scores, and costs of foods purchased. This is the first study, to our knowledge, to propose a method for empirically evaluating the alignment of food procurement data with EAT-*Lancet* planetary health targets across 13 food categories. Given the international focus on leveraging institutional procurement to support a transition to healthy diets from sustainable food systems, analyses like the one undertaken in this study can be useful in evaluating the alignment of food procurement with the EAT-*Lancet* recommendations and for estimating the nutritional, environmental, and cost implications of plant-forward food purchasing shifts. While this study focused on food procurement data from university dining programs, the methods utilized would be applicable in other contexts and could be applied to diverse sets of food procurement data from other institutional settings, consumer food environments, or cities/municipalities. This method of food procurement data analysis could also be useful in the future for setting purchasing targets, monitoring changes in food procurement over time, and evaluating the impacts of healthy and sustainable food policies or behavioral interventions in institutional food service settings.

Food procurement patterns in this sample of U.S. universities exceed EAT-*Lancet* planetary health targets, most notably for beef, pork, and poultry, while failing to achieve targets for legumes and nuts, vegetables, and whole grains. These trends are consistent with the EAT-*Lancet* Commission’s assessment of North American diets [1] as well as a small number of other U.S.-based studies that evaluated the U.S. Dietary Guidelines for Americans and U.S. National School Lunch program menus, both of which failed to achieve EAT-Lancet targets for these same food categories [7,34]. This is likely demonstrative of dietary norms in the U.S. that promote higher meat consumption and lower consumption of healthy plant proteins compared to EAT-*Lancet* recommendations [35]. Future studies should explore whether food procurement data from other contexts, such as K-12 schools, hospitals, and worksites in the U.S., mirror these same trends. While beyond the scope of this study, it would be worthwhile to examine similarities in the composition of food procurement, menus, and food consumption across the meat and healthy plant-food categories within specific food environments (such as universities) to examine correlations across these dimensions. This may also be useful to determine whether food procurement data could be used as a proxy to estimate food consumption within these institutional populations, as consumption data is notoriously difficult and cumbersome to obtain while many institutional food programs have procurement data readily available.

It is evident that significant plant-forward shifts are required to align university food purchases with the EAT-*Lancet* planetary health guidelines. The demonstrated need to increase the prominence of whole, minimally processed plant foods like vegetables, beans and whole grains is well aligned with nutrition priorities that have been previously identified in university settings [11,18]. Given the universities in this study are affiliated with the MCURC, an organization of universities that are uniquely focused on implementing plant-forward initiatives, and that among that subsample of universities, these 19 self-selected to participate in this study, it is possible that a more representative cohort of universities in the U.S. would be even further from the EAT-*Lancet* planetary health targets for these food groups. Nevertheless, there are ample opportunities to implement interventions across university procurement, dining hall menus, and the campus food environment that could move university procurement toward EAT-*Lancet* targets [36].

This study demonstrated that re-aligning food purchases to meet the planetary health targets for all EAT-*Lancet* food categories resulted in significant improvements for the environment, as measured by reductions in food-related GHG emissions and nutritional quality, as measured by HEI scores. These concurrent benefits were consistent across all universities despite differences in the degree of improvement in the GHG emissions and HEI scores between universities. This is consistent with benefits for both GHG emissions reductions and nutritional outcomes that have been documented in studies examining transitions towards the EAT-*Lancet* planetary health diet at individual and global levels [1,37,38]. These results demonstrate an opportunity to highlight the health and environmental co-benefits of plant-forward procurement shifts. 

This study also evaluated the changes in food cost between current food purchases and EAT-*Lancet*-aligned procurement scenarios with a smaller subset of university dining programs that provided food cost data. While the data on food costs was limited to a small number of university dining programs, the majority of those evaluated saw food cost savings in an EAT-*Lancet*-aligned procurement scenario. This finding deserves further exploration in future studies, as cost savings may be an enticing incentive for university dining programs to implement plant-forward shifts in their food purchasing. More research is needed to understand why some universities may see greater food cost reductions than others in order to identify strategies for implementing plant-forward initiatives as a cost-savings strategy. It may also be valuable to explore potential tradeoffs that coincide with cost reductions, for instance, differences in labor required to prepare plant foods (such as beans/legumes, whole grains, and vegetables) compared to meat and poultry.

Using the EAT-*Lancet* planetary health diet as a framework to guide procurement strategies and targets may offer advantages over existing procurement initiatives that focus solely on reducing GHG emissions [17,21]. Focusing on reducing food-related GHG emissions without safeguards or strategies for also addressing the nutritional quality of foods purchased may result in tradeoffs between environmental and nutritional outcomes. It is detrimental from a human health perspective to reduce food-related GHG emissions by displacing meat with plant-based foods that may not offer substantial nutritional benefits, such as ultra-processed meat substitutes, refined grains, sugars, or potatoes [39]. Measuring changes in the contributions of different food categories to the total procurement mix can help safeguard against procurement shifts that reduce the nutritional quality of the procurement portfolio in favor of reducing GHG emissions. Furthermore, evaluating food procurement across the EAT-*Lancet* food categories enables institutional food programs to set category-specific goals that represent the plant-forward shifts they want to incentivize. An example of this would be committing to shifting a percentage of meat purchases towards beans, legumes, and nuts/seeds. Such a commitment would encourage shifts across multiple food categories towards EAT-*Lancet* planetary health targets, yielding reductions in food-related GHG emissions and improvements in nutritional quality.

### Limitations

There are multiple limitations worth noting in this study. While the methods utilized in this paper are aligned with previous assessments of university food procurement data [16,24], the food categorization process required a number of assumptions regarding the composition of foods. The food categorization strategy used in this study was efficient and worked well for commodity foods that fall into a single category but is less accurate for processed and prepared foods that are composed of many food ingredients. While a lack of recipes and ingredient information in the procurement data necessitated such an approach, it is likely that the composition and food category attributions of more complex and highly processed foods were at times over- or under-estimated. This limitation is particularly relevant when looking at changes to HEI scores, which were based on proxy nutrition data representing entire food categories in lieu of having actual nutrition data for all foods purchased. This was a necessary limitation given that the university dining programs in this study did not have comprehensive nutrition data for all foods purchased; however, future studies could examine how well this approach mirrors the nutritional profile using more robust nutrition data. Additionally, the emissions factors used for calculating food-related GHG emissions represented aggregated data for North America, which does not account for nuances in sourcing or production practices at the product level that could be obtained with more accuracy from life cycle assessments (LCAs). However, data on food source or affiliated production practices were not available at the item level and such assessments were beyond the scope of this study. For the nutrition calculations used to generate HEI scores, the estimations for oils, salt, and sugar in the procurement portfolio are likely underestimated since these ingredients make up smaller proportions of mixed foods and are more frequently found in processed food items. Additionally, the EAT-*Lancet* planetary health diet encompasses only whole commodity foods and does not address food processing, so this aspect was not reflected in HEI score calculations. The omission of food processing considerations in the EAT-*Lancet* planetary health diet has been a frequent critique of this framework [40]. Finally, this is a small sample of self-operated universities from an organization (MCURC) that is very focused on promoting plant-forward initiatives in university dining. Small universities (<5000 students) were not represented in our sample despite making up 9.8% of universities in the MCURC, which may limit the generalizability of these results to that group. Regardless, it is likely that the universities in our sample are more aligned with the EAT-*Lancet* planetary health diet than a more representative sample of universities that are less focused and progressed in their efforts to advance plant-forward purchasing. Future studies should aim to assess data from universities beyond the MCURC, as well as contract operated dining programs that operate within universities and other institutional environments.

## 5. Conclusions

This study demonstrates that food purchases in this sample of U.S. universities are not currently aligned with the EAT-*Lancet* planetary health diet. However, results indicate that universities could realize concurrent benefits for reducing food-related GHG emissions, improving nutritional indicators, and lowering food costs by transitioning towards plant-forward procurement practices that are aligned with the EAT-*Lancet* planetary health diet. Plant-forward procurement shifts that displace substantial amounts of meat with whole, minimally processed plant foods could have positive implications for enhancing the healthfulness and sustainability of university food environments and improving the eating behaviors of students. The procurement metrics described in this study could be helpful in setting purchasing targets and monitoring progress towards alignment with EAT-*Lancet* planetary health targets, as well as evaluating potential nutrition and environmental impacts for foods purchased in universities and other institutional settings.

## Figures and Tables

**Figure 1 ijerph-21-00945-f001:**
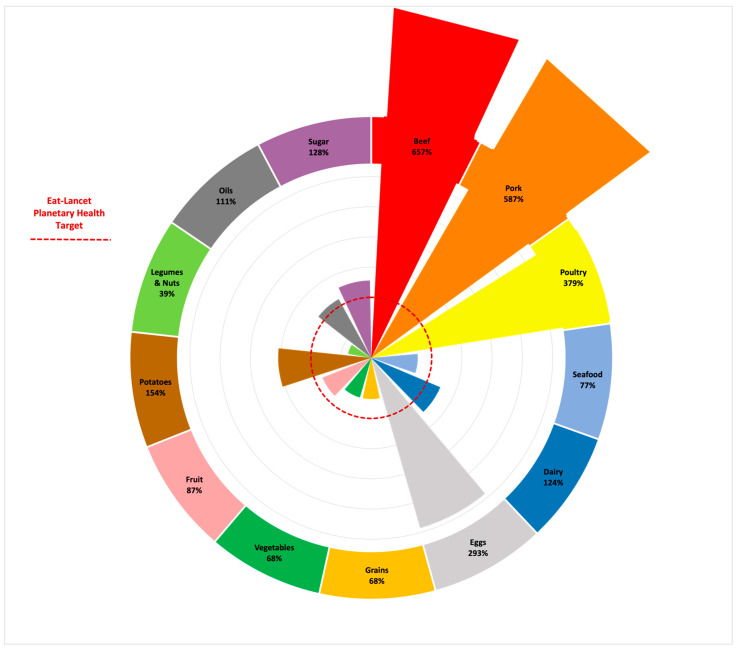
Average Deviation (%) from EAT-*Lancet* Planetary Health Target Across 13 Food Categories for Foods Procured in 2022 by a Sample of 19 U.S. Universities. Note: The average deviation (%) from the planetary health target was determined by comparing the composition of foods purchased in 2022 in the 19-university sample with planetary health targets for each food category in the EAT-*Lancet* planetary health diet [1]. The percent contribution of a food category to the total mix of foods (by weight) was divided by the “ideal” percentage of that food category in the planetary health diet to arrive at the % deviation indicated under each food category. The pie pieces represent the % deviation visually in comparison to the planetary health target (indicated by the red dotted line in the figure). Mean values > 100% (extending beyond the red dotted line) indicate the universities exceeded the EAT-*Lancet* planetary health target for a given food category, and mean values < 100% (falling within the red dotted line) indicate the universities are below the EAT-*Lancet* planetary health target for a given food category.

**Table 1 ijerph-21-00945-t001:** Descriptive Characteristics of Participating Universities Compared to All MCURC Member Universities *.

	Universities in Study (*n* = 19)	Eligible MCURC Universities * (*n* = 41)
University Size **		
Small (<5000 students)	0.0% (*n* = 0)	9.8% (*n* = 4)
Medium (5–15,000 students)	21.1% (*n* = 4)	19.5% (*n* = 8)
Large (>15,000 students)	78.9% (*n* = 15)	70.7% (*n* = 29)
Region ^+^		
North	31.6% (*n* = 6)	36.6% (*n* = 15)
Midwest	10.5% (*n* = 2)	12.2% (*n* = 5)
South	10.5% (*n* = 2)	7.3% (*n* = 3)
West	47.4% (*n* = 9)	43.9% (*n* = 18)
Public/Private ^+^		
Private	47.4% (*n* = 9)	39.0% (*n* = 16)
Public	52.6% (*n* = 10)	61.0% (*n* = 25)
% Female Students ^+^	Mean = 52.0%	Mean = 53.0%
% Minority Students ^+^	Mean = 53.0%	Mean = 45.0%
% Pell Grant Students ^+^	Mean = 22.0%	Mean = 23.0%

* Only MCURC universities in the U.S. with affiliated dining programs are included. ** Carnegie Classification [22]. ^+^ U.S. News and World Report [23].

**Table 2 ijerph-21-00945-t002:** Contribution of Food Categories to the Total Mix of Foods in EAT-*Lancet* Planetary Health Diet Compared to Foods Purchased in 2022 for a Sample of 19 U.S. Universities.

EAT-*Lancet* Food Category	EAT-*Lancet* Planetary Health Target (% by Weight)	Average Composition of Foods Purchased in 2022 by 19 Universities (% by Weight)	Range (*n* = 19)	Standard Deviation (*n* = 19)
Beef	0.53%	3.48%	2.11–6.13%	1.02%
Pork	0.53%	3.11%	2.05–6.74%	1.13%
Poultry	2.20%	8.33%	4.53–13.05%	2.41%
Seafood	2.12%	1.63%	0.20–4.96%	1.08%
Dairy *	18.96%	23.56%	11.48–38.34%	6.75%
Eggs	0.99%	2.90%	1.18–4.13%	0.65%
Grains	17.59%	12.00%	6.86–24.39%	3.94%
Vegetables	22.75%	15.42%	10.14–22.66%	3.80%
Fruits	15.17%	13.19%	6.80–24.58%	4.96%
Potatoes	3.79%	5.85%	2.48–8.09%	1.76%
Legumes & Nuts	9.48%	3.65%	0.72–6.89%	1.37%
Oils	3.55%	3.94%	1.75–14.54%	2.74%
Sugars	2.35%	3.00%	1.63–5.23%	1.20%

Note: Percentages refer to the proportion of total foods (by weight) for each food category (i.e., the “Procurement Portfolio”). EAT-*Lancet* Planetary health target determined from the contribution of each food component to the total mix of foods (by weight) in the EAT-*Lancet* planetary health diet. Procurement portfolio composition represents the average contribution of each food category (% by weight) to the total mix of foods in the 19-university sample. * Dairy is based on whole-milk equivalents as described in the EAT-*Lancet* planetary health diet. Weight of solid dairy foods (i.e., cheese, butter) was converted to equivalent weight in milk based on the USDA My Plate serving [28].

**Table 3 ijerph-21-00945-t003:** Change in GHG Emissions and HEI Scores in Current Procurement Portfolio versus EAT-*Lancet*-Aligned Scenario for 19 Universities.

University	GHG Emissions (kg CO_2_e) per kg of Food Purchased			HEI Scores (Out of 100)		
	Current Procurement Portfolio	EAT-*Lancet*-Aligned Scenario	Change (%)	Current Procurement Portfolio	EAT-*Lancet*-Aligned Scenario	Change (Points)
1	3.65	1.87	−48.9%	64.48	85.99	+21.51
2	3.11	1.86	−40.2%	66.41	85.73	+19.31
3	3.73	1.84	−50.7%	53.49	85.89	+32.41
4	3.19	1.85	−42.1%	68.66	84.31	+15.64
5	2.75	1.84	−33.2%	75.39	80.60	+5.21
6	3.66	1.84	−49.7%	47.63	73.93	+26.30
7	3.06	1.85	−39.6%	66.17	81.99	+15.82
8	3.15	1.86	−41.0%	70.14	84.32	+14.19
9	3.64	1.85	−49.1%	67.26	85.76	+18.50
10	3.83	1.94	−49.4%	66.15	86.77	+20.62
11	3.92	1.92	−51.1%	58.35	84.60	+26.25
12	3.54	1.85	−47.7%	58.48	86.37	+27.89
13	2.94	1.89	−35.6%	63.00	79.13	+16.13
14	3.31	1.90	−42.4%	65.31	85.01	+19.71
15	3.84	1.89	−50.8%	69.59	82.85	+13.26
16	4.57	1.88	−58.9%	63.58	82.49	+18.91
17	3.26	1.82	−44.3%	61.24	80.85	+19.61
18	4.06	1.86	−54.2%	63.39	81.63	+18.23
19	2.79	1.94	−30.0%	60.13	84.90	+24.77
Average	3.47	1.87	−46.1%	63.62	83.32	+19.70

Note: As total GHG emissions can vary substantially based on university size, total emissions divided by the total weight (kg) of foods purchased was used to compare food-related GHG emissions across universities.

## Data Availability

Restrictions apply to the availability of these data. Data were obtained from participating universities under the condition of anonymity and are available from the corresponding author with the permission of the participating institutions.

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
