# Peer review of "Evaluating Food Procurement against the EAT-Lancet Planetary Health Diet in a Sample of U.S. Universities"

_ijerph, 2024, doi:10.3390/ijerph21070945_

Round 1

Reviewer 1 Report

Comments and Suggestions for Authors

Overall, this was an interesting, well researched and thorough paper. I have a few minor suggestions, listed below. 

Line 44 – This is pedantic but I would add a line that stresses that the EAT-Lancet diet is not vegan, and does not propose veganism. It recommends a significant reduction in the (scientifically proven to be) unhealthy levels of meat, particularly red meat, common in the diets of more affluent countries. Detractors often jump to criticisms that a vegan diet may not provide sufficient nutrients, etc. I think it is worth acknowledging and heading off these potential criticisms as early as possible. 

Line 180 – please state if the TASTE food tool captures the contribution of food miles to the overall value of GHG emissions, as well as just the type of food. Does it differentiate between different lifestock systems such as caged-raised broiler chickens and free-range chickens, for example. This is made clear later but I think it would be useful to give a quick acknowledgement here, too. 

Line 496(ish) Somewhere around here I would be inclined to add a line acknowledging the danger of moving from meat to ultraprocessed meat substitutes which can be equally bad for the environment and health. Acknowledging that the shift away from meat should be to wholefood plant alternatives is worth a little space. 

Author Response

Comments 1: Line 44 – This is pedantic but I would add a line that stresses that the EAT-Lancet diet is not vegan, and does not propose veganism. It recommends a significant reduction in the (scientifically proven to be) unhealthy levels of meat, particularly red meat, common in the diets of more affluent countries. Detractors often jump to criticisms that a vegan diet may not provide sufficient nutrients, etc. I think it is worth acknowledging and heading off these potential criticisms as early as possible.

Response 1: Thank you for making this point. We have addressed this comment by adding additional information on page 1, paragraph 2, lines 42-53:

“The EAT-Lancet planetary health diet aims to improve dietary health and reduce food-related greenhouse gas (GHG) emissions among other environmental impacts by promoting a plant-forward diet that emphasizes whole and minimally processed grains, vegetables, fruit, legumes, and nuts while limiting meat and dairy [1]. Plant-forward diets are not vegan, and the planetary health diet accommodates an omnivorous eating pattern that is intended to be adaptable across different cultures and contexts. However, it recommends substantial reductions in meat consumption, particularly for affluent countries that are heavy consumers.

Comments 2: Line 180 – please state if the TASTE food tool captures the contribution of food miles to the overall value of GHG emissions, as well as just the type of food. Does it differentiate between different lifestock systems such as caged-raised broiler chickens and free-range chickens, for example. This is made clear later but I think it would be useful to give a quick acknowledgement here, too.

Response 2: We appreciate the opportunity to clarify this point, as the TASTE food tool was not used to calculate emissions from the foods purchased, only to categorize the itemized foods for analysis. We have now noted that we utilized the food categorization function of the TASTE food tool on page 3, paragraph 5, line 158:

The procurement data was first run through the food categorization function of the Automated Scope 3 Tool for Tracking Emissions from Food (TASTE Food), a python-based procurement analysis tool that was previously developed and validated with university food procurement data [24,25].  

Comments 3: Line 496(ish) Somewhere around here I would be inclined to add a line acknowledging the danger of moving from meat to ultraprocessed meat substitutes which can be equally bad for the environment and health. Acknowledging that the shift away from meat should be to wholefood plant alternatives is worth a little space.

Response 3: We agree with this point, and have emphasized this in both the discussion and the conclusion:

Page 12, paragraph 3, line 472: Focusing on reducing food-related GHG emissions without safeguards or strategies for also addressing the nutritional quality of foods purchased may result in tradeoffs between environmental and nutritional outcomes. It is detrimental from a human health perspective to reduce food-related GHG emissions by displacing meat with plant-based foods that do not offer substantial nutritional benefits, such as ultra-processed meat substitutes, refined grains, sugars, or potatoes [39].

Page 13, paragraph 2, lines 609-611: Plant-forward procurement shifts that displace substantial amounts of meat with whole, minimally processed plant-foods could have positive implications for enhancing the healthfulness and sustainability of university food environments and improving the eating behaviors of students.

Reviewer 2 Report

Comments and Suggestions for Authors

Congratulations. Im reviewing this paper for the first time so I can see where changes/amendments have been made. Im abundantly aware of the numerous assumptions you have made - and while the results may be imperfect - by utilising the work of others your findings are sound and of interest. I wholeheartedly agree with the potential bias present in your sample - these universities are already moving to plant based diets - hence within the broader population I would also expect to see an even greater deviation from the proposed planetary diets. 

Author Response

Comments 1:  Congratulations. Im reviewing this paper for the first time so I can see where changes/amendments have been made. Im abundantly aware of the numerous assumptions you have made - and while the results may be imperfect - by utilising the work of others your findings are sound and of interest. I wholeheartedly agree with the potential bias present in your sample - these universities are already moving to plant based diets - hence within the broader population I would also expect to see an even greater deviation from the proposed planetary diets.

Response 1: Thank you for providing this positive feedback. We appreciate your review and perspectives on the manuscript.

Reviewer 3 Report

Comments and Suggestions for Authors

Manuscript ID: ijerph-3087172

Title: Evaluating Food Procurement Against EAT-Lancet Planetary Health Guidelines in a Sample of U.S. Universities

The study provides novel data on an important topic. The manuscript needs some modifications so that it could be better than before. It would be helpful if the authors would consider the following points:

- Lines 42-43: There is a repetition of this word "whole" in this sentence

- Write the aim of the study at the end of the introduction and not at the beginning of the Materials and Methods

- Design a FIGURE that shows a general description of Materials and Methods in the manuscript, making it easier for the reader to understand

- Line 205: " % sugar " ?

- In Table 1: Universities in Study in category of Small (<5,000 students) is zero. Does this zero not affect the efficiency of the study and the validity of the statistical analysis?

- In Figure 1: The words and percentages within Figure 1 are unclear. Please explain it better than that.

- The discussion section is based on only seven references, and this is a small number to produce a strong discussion, so this section must be supported by some modern references.

- Discussion seems to be poor, didn't give good explanations of the results obtained. I think that it must be really improved.

- It is preferable to shorten the "Limitations" section

- Insert the correct format style for journals in the references in the text and references list.

- There are a large number of references that are links, which weakens the strength of the manuscript.

Comments on the Quality of English Language

Minor editing of English language required

Author Response

Comments 1:  Lines 42-43: There is a repetition of this word "whole" in this sentence

Response 1: Thank you for pointing this out. The sentence has been revised to avoid repetition (page 1, paragraph 2, line 43)

“The EAT-Lancet planetary health diet aims to improve dietary health and reduce food-related greenhouse gas (GHG) emissions among other environmental impacts by emphasizing a plant-forward diet in which whole and minimally processed grains, vegetables, fruit, legumes, and nuts comprise a greater proportion of foods consumed than meat and dairy [5,6].”  

Comments 2: Write the aim of the study at the end of the introduction and not at the beginning of the Materials and Methods

Response 2: Thank you for this suggestion, the aim of the study has been relocated to the end of the introduction (page 3, paragraph 1, lines 103-106).

Comments 3: Design a FIGURE that shows a general description of Materials and Methods in the manuscript, making it easier for the reader to understand

Response 3: We have incorporated this suggestion and now include a process diagram for the materials and methods in this study as a supplemental figure (Figure S1) in the appendix (page 14, line 538).

Comments 4: Line 205: " % sugar " ?

Response 4: We appreciate the opportunity to clarify this point. This sentence has been revised:

For sugar-sweetened grain products, dairy, and beverages, the percentage of the product’s weight that was comprised of sugar was approximated using USDA nutritional data from reference foods and attributed to the sugar category, while the remaining weight was distributed across other relevant food categories (i.e. grains, dairy, liquids).

Comments 5: In Table 1: Universities in Study in category of Small (<5,000 students) is zero. Does this zero not affect the efficiency of the study and the validity of the statistical analysis?

Response 5: This is certainly a limitation that affects the generalizability of the results to all universities and is an opportunity for future research. This has been highlighted in the limitations section (page 13, paragraph 1, lines 504-506):

Small universities (<5,000 students) were not represented in our sample despite making up 9.8% of universities in the MCURC, which may limit the generalizability of these results to that group.

Comments 6: In Figure 1: The words and percentages within Figure 1 are unclear. Please explain it better than that.

Response 6: We appreciate the opportunity to better explain this figure. The note underneath the figure has been updated to offer more clarity (page 9, lines 355-364):

Note: The average deviation (%) from the planetary health target was determined by comparing the composition of foods purchased in 2022 in the 19-university sample with planetary health targets for each food category in the EAT-Lancet planetary health diet [1]. The percent contribution of a food category to the total mix of foods (by weight) was divided by the “ideal” percentage of that food category in the planetary health diet to arrive at the % deviation indicated under each food category. The pie pieces represent the % deviation visually in comparison to the planetary health target (indicated by the red dotted line in the figure). Mean values > 100% (extending beyond the red dotted line) indicate the universities exceeded the EAT-Lancet planetary health target for a given food category and mean values < 100% (falling within the red dotted line) indicate the universities are below the EAT-Lancet planetary health target for a given food category.

Comments 7: The discussion section is based on only seven references, and this is a small number to produce a strong discussion, so this section must be supported by some modern references.

Response 7: The discussion section has been revised with the addition of six more references to contextualize our results from studies published in 2019 or later (see pages 11-12).

Comments 8: Discussion seems to be poor, didn't give good explanations of the results obtained. I think that it must be really improved.

Response 8: Thank you for this feedback. As mentioned above, the discussion section has been revised to better contextualize the results of this study, including references to additional literature to which we compare our results, and additional discussion of possible explanations for our findings (see pages 11-12).

Comments 9: It is preferable to shorten the "Limitations" section

Response 9: We appreciate this perspective but think it is important to discuss the various limitations of this study so that readers understand what they are. Therefore, we prefer not to shorten this section.

Comments 10: Insert the correct format style for journals in the references in the text and references list.

Response 10: We have confirmed that the style in our paper is consistent with IJERPH standards.

Comments 11: There are a large number of references that are links, which weakens the strength of the manuscript.

Response 11: Revisions to the manuscript have resulted in fewer references that are links. However, we feel in this area of work there is value in citing resources outside the peer reviewed literature, and the references we cite are from high quality sources that we do not believe negatively impact the quality of the manuscript.